# The Role of Tailored Public Health Messaging to Young Adults during COVID-19: "There's a lot of ambiguity around what it means to be safe"

Tina Cheng[1,2], Braxtyn Horbay[3,4], Rochelle Nocos[2,5], Lesley Lutes[3,4], Scott A. Lear[5,6]*

1 School of Population and Public Health, University of British Columbia, Vancouver, British Columbia, Canada, 2 Community Health Research Team, Vancouver, British Columbia, Canada, 3 Department of Psychology, University of British Columbia, Okanagan, British Columbia, Canada, 4 Centre of Obesity and Well-being Research Excellence, University of British Columbia, Okanagan, British Columbia, Canada, 5 Faculty of Health Sciences, Simon Fraser University, Vancouver, British Columbia, Canada, 6 Division of Cardiology, Providence Health Care, Vancouver, British Columbia, Canada

* slear@providencehealth.bc.ca

**Data Availability Statement:** The data used in this study pertains to transcripts from interviews, some with personal experiences. Therefore, we are not able to share the transcripts Participants were not

## Abstract

The COVID-19 global incidence rate among young adults (age 19–40) drastically increased since summer 2020, and young adults were often portrayed by popular media as the "main spreader" of the pandemic. However, young adults faced unique challenges during the pandemic due to working in high-risk, low-paying essential service occupations, as well as having higher levels of financial insecurity and mental burden. This qualitative study aims to examine the attitudes and perceptions of health orders of young adults to better inform public health messaging to reach this demographic and increase compliance to public health orders. A total of 50 young adults residing in British Columbia, Canada, were recruited to participate in focus group in groups of four to six. Focus group discussions were conducted via teleconferencing. Thematic analysis revealed four major themes: 1) risks of contracting the disease, 2) the perceived impact of COVID-19, 3) responsibility of institutions, 4) and effective public health messaging. Contrary to existing literature, our findings suggest young adults feel highly responsible for protecting themselves and others. They face a higher risk of depression and anxiety compared to other age groups, especially when they take on multiple social roles such as caregivers and parents. Our findings suggest young adults face confusion due to inconsistent messaging and are not reached due to the ineffectiveness of existing strategies. We recommend using evidence-based strategies proven to promote behaviour change to address the barriers identified by young adults through tailoring public health messages, specifically by using positive messaging, messaging that considers the context of the intended audiences, and utilizing digital platforms to facilitate two-way communication.

informed or requested to allow for the data to be shared publicly. The Simon Fraser University Office of Research Ethics can be contacted at 778-782-3447 (https://www.sfu.ca/senate/senate-committees/reb.html).

**Funding:** The author(s) received no specific funding for this work.

**Competing interests:** The authors have declared that no competing interests exist.

## Introduction

When the COVID-19 pandemic first began in early 2020, the focus of public health measures and healthcare systems was to protect seniors and those with compromised immune systems. This focus was both prudent and understandable given age was, and still is, the strongest risk factor for COVID-19 mortality [1]. While early in the pandemic, cases were highest amongst older adults over the age of 65 years old, cases in young adults soon outpaced those of any other age group [2]. In the summer of 2020, the World Health Organization issued warnings citing young adults as the main "spreaders of the virus", as the COVID-19 incidence rate among the young adult age group was rapidly increasing around the world [3, 4]. In the United States, it was reported young adults between 20–49 years old accounted for more than 70% of the spread in 2020 [5]. In addition, this age group will likely receive their vaccinations last as countries vaccinate their population based on age, yet they have the highest vaccine hesitancy [6].

Young adults also face unique challenges in the COVID-19 pandemic due to public health orders restricting social interactions at a time of crucial social development and independence, working in high-risk, low-pay essential service jobs, employment in occupations without medical benefits, and financial insecurity [7, 8]. While others may have the option to safely stay at home due to the flexibility of their occupations, many young adults are essential workers and do not have this same option. In addition to the low-paying jobs that do not provide sick leave, young adults in such public-facing occupations are at a higher risk of COVID-19 exposure and infection. However, studies have argued the lower perceived risk and worry regarding COVID-19 in young adults may have resulted in lower compliance with public health orders such as restrictions on social gatherings and mask wearing compared to older adults [9–11]. Moreover, mass media stories and public health messages have resorted to blaming and shaming for individual decisions on health behaviours despite evidence indicating this as ineffective [5, 12, 13].

Despite the high number of cases and concern of adherence to public health messages, few jurisdictions have implemented public health messaging targeting this age group. There is limited understanding of young adult perceptions on COVID-19 and effective public health messaging towards youth for communicable diseases [14, 15]. Therefore, the current study sought to explore the experiences and perceptions of young adults aged 18–40 in British Columbia (BC), Canada relating to the COVID-19 pandemic to best inform public health messaging.

## Method

Participants were recruited via existing research networks, social media (Twitter, Instagram), and snowball recruitment from October to November 2020. Participants were eligible to participate if they were between 18 and 40 years of age, had resided in B.C. since March 1, 2020, had access to a device capable of an online conferencing software either by computer, tablet or phone and were able to provide informed consent. The inclusion criteria that participants had to reside in B.C. since March 1, 2020, ensured the participant resided in B.C. since the first provincial lockdown on March 17th, 2020. Over 200 potential participants showed interest to participate, but seven did not meet the inclusion criteria. We strived for representation across gender, geography, employment status and education. Following approval from the Simon Fraser University Research Ethics Boards, 50 participants were recruited to participate in the study.

Participants gave verbal or written informed consent prior to enrollment in the study and were scheduled into groups of 4–6 to participate in a single, semi-structured, open-ended focus group audio-recorded via Zoom video conferencing. The focus group discussion

methodology was chosen for its strength in providing access to in-depth understanding of shared social meanings or norms. Through focus group discussions, we aimed to explore the natural interactions participants might have in their daily lives and how these interactions may influence their perception of existing health messaging. Each focus group was attended by one moderator and one notetaker from the research team. The focus groups were conducted in English and lasted between 60–90 minutes, depending on the participants' engagement in the discussion. Each participant was identified in the transcript by the focus group they participated in and the gender of the participant. The research team facilitated each focus group with a semi-structured question guide (S1 Appendix) that inquired about participant experiences and perceptions of current public health messaging on COVID-19. All audio recordings were transcribed verbatim.

## Focus group discussion analysis

Qualitative analysis of the focus group transcripts was performed by two experienced qualitative researchers. Using NVivo qualitative thematic analysis software $12^{TM}$ (QSR International PTY Ltd, Melbourne, Australia), findings were compiled on the verbatim transcription of the transcripts following the six phases of thematic analysis as outlined by Braun and Clarke [16, 17]. The phases summarized in Table 1 were completed

Themes were inductively identified from the raw data without any predetermined classification. The researchers first generated rough codebooks to guide the coding process and the codebooks were refined as the coding process went on. This coding process was interactive and reflective with movement between phases and two experienced qualitative data analysts of the research team working in collaboration, minimizing bias in the interpretation of the interview data. Discrepancies among codes that emerged throughout were reviewed and merged with other suitable themes after group discussions until consensus was reached. We closely followed the consolidated criteria for reporting qualitative research (COREQ) checklist to guide and report our findings [18].

## Results

### Participant demographics

A total of 50 participants within the age range of 18–40 ($M$ = 29 years, $SD$ = 6.7) participated in the focus groups. Almost half of the participants were women and the other half men with

**Table 1. Phases of thematic analysis adapted from Braun and Clarke.**

| Phases | Description of Process |
|---|---|
| 1. Data familiarization | Familiarized ourselves with the data by transcribing and reviewing the recordings into written form. Transcripts were read and re-read. |
| 2. Generating initial codes | Researchers first generated initial codes individually via open coding. Coding was completed on a line-by-line basis where codes were assigned to relevant words, sentences, and paragraphs. Emerging patterns and interesting features were noted. |
| 3. Searching for themes | Excerpts from each code were re-read. Themes were generated by organizing initial codes across all eight focus group transcripts. |
| 4. Reviewing themes | Researchers collectively reviewed the themes to reach a consensus through a series of group discussions. Extracts from each code were reviewed to check whether they are represented by the themes. |
| 5. Defining and describing themes | Researchers defined and named the themes to ensure accurate representation of central ideas and concepts within the data. Ongoing analysis was performed to refine the specifics of each theme and checked that each theme captured unique features in relation to the overall research question. |
| 6. Producing the report | Produced a written report of the results |

**Table 2. Participant demographics.**

| | 18–29 years old | 30–40 years old | Total |
| --- | --- | --- | --- |
| | N = 27 | N = 23 | N = 50 |
| Gender | | | |
| Men | 16 (59%) | 8 (35%) | 24 |
| Women | 11 (41%) | 12 (52%) | 23 |
| Non-binary | 0 (0%) | 3 (13%) | 3 |
| Urban Residence | 23 (85%) | 18 (78%) | 41 |
| Employed | 16 (59%) | 17 (74%) | 33 |
| Post-Secondary Education | 27 (100%) | 23 (100%) | 50 |

three non-binary participants Table 2. All participants had completed a high school education and were either currently enrolled in a post-secondary education program or had already completed one, and 82% of participants resided in an urban area.

Qualitative thematic analysis revealed four domains of themes within the data: risk of disease contraction, perceived sociocultural impact of COVID-19, role of institutions, and effective public health messaging Table 3. Each domain has a number of subthemes that fall within the overarching theme. Each theme was reviewed in detail for additional elucidation and potential guidance on next steps regarding messaging and recommendations to government officials.

**I: Risk of disease contraction.** While most participants believed they have been practicing COVID-19 safety measures as promoted by public health, such as keep small social circles and avoid unessential meetings, participants acknowledged that compliance to safety measures lies on a spectrum. One participant explained:

> "For me I think it's very concerning that a lot of people have a different mindset or different information that they're getting. The fact that it's dividing a lot of people within what is right and wrong is very concerning." (FG4 F)

Participants acknowledged the added anxiety from constantly assessing the risk of their daily behaviours. As individual perceptions of safety vary, participants expressed their concerns when their threshold of safety was different from others. Power dynamics were explored

**Table 3. Overview of themes.**

I: Risk of Disease Contraction: participant's perception of actions that would increase their risk of contracting the virus

II: Impact of COVID 19
  • Epidemiology: participant's perceptions on features and information of the virus
  • Individual: both biological and psychological impact the pandemic has on an individual
  • Societal: perceived short- and long- term impact of the virus on society, with a focus on economic pressure faced by local small businesses

III: Role of institutions
  • Media: participant perception on the function and purpose of media during the pandemic
  • Government: participant perception on the responsibility and action of the government during the pandemic
  • Individual: responsibility an individual should take to lower transmission during the pandemic

IV: Effective Public Health Messaging
  • Information Seeking and Receiving: where young adults actively seek public health information and where they receive such information passively
  • Suggestions for Effective Messaging: concerns and questions participants had regarding current messaging lead to suggestions for future messaging

when participants had to weigh the risk according to their own safety standards and reject social invitations, or bear sacrifices in their lives. In particular, majority of participants felt they unconsciously judge others based on their compliance to public health orders.

> "I think that my friends are taking it seriously but I don't feel like they are meeting the standards that I would take, so that is a little bit awkward." (FG1 F)

**II: Impact of COVID-19.** Participants recognized the depth of impact COVID-19 has had on epidemiological, individual, and societal levels.

*Epidemiology.* Participants gave uniform answers on knowledge of the virus origin, transmission, health consequences and treatments. They discussed the virus both from a immunological and geographic perspectives where one stated:

> "I guess some more information we could add is like the origins, where it started in China and it travelled across the world due to transportation not being shut down quickly enough." (FG3 M)

Participants also believed the virus was transmitted through small particles that were either airborne or through droplets. A few participants worried about how asymptomatic individuals could spread the disease without knowing they had it because of the long incubation period in which individuals do not develop symptoms. Many participants spoke on their knowledge of the virus:

> "It's in the same family of diseases as SARS and MERS, it mainly attacks the respiratory system, and it's spread through droplets predominantly." (FG2 F).

> "It is extremely contagious and requires people to protect themselves with facemasks or other like PPE." (FG1 F)

All participants recognized there are no existing treatments or cures for COVID-19 and acknowledged the difficulties faced by public health in finding a treatment or vaccine. However, participants agreed vaccines are deemed necessary for a safe return to normal lives, yet many still worried about the rapid development of vaccines. One participant explained:

> "I just hope that when the vaccine is actually released that it's gone through rigorous testing and its actually proven to be safe because I feel that the timeline right now like for vaccines to be developed is rushed." (FG1 F)

*Individual.* Participants expressed their concerns for long-term physical and mental health consequences, fear of contracting the virus as well as the stigma around contracting the virus. Lifestyle adjustments such as making decisions on safety, financial burdens, as well as the importance of family were also common topics within the impact of COVID-19 theme. One participant explained:

> "I don't drive, and I take public transit. So I just wonder how many active cases there are, and think, like, what are the consequences of doing this? What is the likelihood that I'll get sick? And what are the consequences of not doing this, like what are consequences to my mental health, my motivation?" (FG6 M)

Another participant described the sudden loss of financial income:

"So, the entire month of April [2020], I guess about six weeks, actually, my husband and I were both without income. And he's a physiotherapist and he had recently opened his own clinic, so that sucked a lot. Anyway, yeah, so went from pretty decent income to zero over-night." (FG3 F)

Participants also mentioned their frustrations with the federal financial supports, explaining the inability to fit into the support criteria. While many also reported a loss of financial income during COVID-19, most participants discussed how not being able to take care of, see, and connect with their family members being the most devastating part of the pandemic. One participant explained:

"I think I, just echoing everybody, it's definitely been the, the family interaction and how that's changed." (FG6 F)

"I'm not as concerned about myself, but like, I have elderly grandparents and my parents are older, and so, like, I know for me my biggest fear is, like, contracting it and not knowing and then giving it to other people and getting them sick." (FG7 F).

*Societal.* There was an overall concern of the impact that COVID-19 will cause on the economy focusing on small businesses. Many believed more aid from the government should be available to support both the individual and other small outlets. One participant expressed:

"I think, for me what, what really, like stresses me out with COVID-19 is how it's impacting our small businesses and how so many people are unable to get work or have lost their jobs. And just, just on a, purely, like, the economic fallout of it has been very stressful to me." (FG 5 F)

**III: Role of institutions.** This theme focused on the three major institutions that participants identified to have clear responsibilities in the COVID-19 outbreak response: the government, the media, and the individual.

*Government.* Participants believed it is the government's responsibility to mandate and enforce health orders set out by public health officials. More importantly, participants believed there should be more positively framed messaging coming from the government that encompass more emotional appeals to adhere to restrictions. One participant explained:

"Instead of the government just kind of keep rolling the same ball, let's pick it up and throw it through the hoop. Like, have something, something different, and I think people would be a little bit more responsive to being like, okay, if we just buckle down and we, we work on this, you know, we can get through this. I think if the government had a different approach doing that way, I think we'll definitely be a lot more responsive." (FG8 M)

*Media.* Participants recognized media as a very important medium to deliver information. It was noted by participants in the focus groups they remembered seeing reports about people in their age group violating COVID-19 health orders. However, participants also believed that, at times, exaggerated and biased wording was being used by the media to describe the behaviour of young adults. When discussing how young adults were being portrayed on the news, a participant mentioned:

"They are disregarding the fact that young adults are also the bulk of the working population who don't get the choice to go home and work in their home office are these same people. So yes, you know, the, the, there's accuracy in that it's more and more young people are driving infections, but there are more and more young people that have to be out. Have to resume life because they don't have the savings or the luxury to alternate their work schedule." (FG5 M)

The lack of balanced reporting of both positive and negative behaviours of young adults by the media left the participants unsatisfied. On the other hand, there were also concerns expressed on how both the timing and the content of the information being shared on mainstream media and the messages being shared by the government failed to align with one another. While they appreciated that the media must report stories as they happen, these challenges and conflicts ultimately raised questions regarding the motivations, values, and ultimately, the credibility of the news media as a whole.

*Individual.* Participants acknowledged the responsibility of the individual to keep both themselves and others safe, and that only a collective individual effort will stop the pandemic. By staying at home when ill, maintaining physical distancing, and wearing a mask, participants believed practicing non-pharmaceutical interventions are a way to show willingness to protect others.

"I think in theory, nobody would have to take responsibility for themselves if every single person took responsibility for others. However, obviously that's not happening, it's not the case." (FG6 M)

Participants believed each institution shared unique responsibilities to contribute to the control of spread, enforcement of guidelines and punishment for those who do not follow guidelines and the rebuild when the pandemic is declared over. Collectively, they believed that each group owned a shared responsibility, between the government, the media, and the individual, in stopping the spread of COVID-19 and, ultimately, the plan for a safe return to a quality of life which includes both economic productivity, meaningful social engagement, and improved health and well-being.

**IV: Effective public health messaging.**   The theme of effective public health messaging refers to the suggestions made to ultimately improve the effectiveness of the public health messaging surrounding the COVID-19 pandemic, vaccination, and return to meaningful daily interactions. During the focus groups, participants emphasized the importance of information seeking and receiving and identifying the current critiques of public health messaging.

*Information seeking and receiving.* While participants acknowledged social media platforms such as Instagram, Facebook, and TikTok are often not trustworthy, they are the most popular medium young adults use for news, as well as information seeking and receiving on a daily basis. One participant explained:

"Instagram is the one place I am going to go every single day regardless of whether or not I have the energy to look at the news and I do find the government is really slow to advertise messaging on that platform." (FG2 M)

Participants also claimed that after receiving information from social media they often cross reference the information by looking for credible sources through government websites, scientific articles, news networks and health authorities.

*Suggestions for effective messaging.* Most participants voiced their concerns regarding the uncertainties in public health messaging and highlighted the quick changes in public health orders. For example, one participant mentioned:

> "Recently with public health saying to limit ourselves to one household and six safe people I think there's a lot of ambiguity around what it means to be safe." (FG2 F)

Many participants suggested utilizing multiple social media platforms to deliver public health messages and claimed messages would be stronger if presented with emotion and in a positive and creative manner. Responses regarding such suggestions included:

> "It needs to be geared towards the, the base positive feelings that all human beings have, which is the need to connect, which is the need to belong, the need to feel cared about, the need to feel valued." (FG8 M)

While statistical projections should still be provided, participants longed for messages that relay empathy in recognizing people's hardships. Participants also desired for one unifying message that instilled hope for a return to normal life while also motivating the public to adhere to restrictions. Finally, participants wished for contrary projections to the traditional negative bias of young adult behaviour with that of positive role models and the effects to be had if everyone committed to safe practices as a way of communicating a light at the end of a very long and dark tunnel that is the COVID-19 global pandemic.

## Discussion

The current qualitative study examined young adults' perceptions of the COVID-19 pandemic and the effectiveness of current public health messaging. Overall, our participants, contrary to mass media coverage, were highly concerned about the spread of COVID-19 and perceived it as their responsibility to protect themselves and their loved ones. Similarly, recent research found younger adults have a higher perception of risk vulnerability with respect to COVID-19 [14, 19]. Young adults in our study highlighted the difficulties they faced and resiliency they demonstrated in dealing with COVID-19. Furthermore, they expressed concerns and confusion around current public health messaging and provided valuable insights for future effective messaging on COVID-19 as we enter the next phase of the pandemic–vaccination.

### The power of positive messaging

A heightened level of stress and anxiety was noted to be felt consistently through discussion among participants. Similarly, young adults in other studies were also found to be moderately less optimistic about the pandemic's outlook and perceive higher levels of risk vulnerability than older adults, as well as anxiety and depression [14, 19–21]. Previous research also suggests young adults are more easily stressed if health threats are perceived, leading to long-term negative mental health consequences such as anxiety and depression [22–25]. Young adults also have experienced far greater psychological distress symptoms during the pandemic than older generations [26]. Stress may be further exacerbated by lower savings, less occupation stability, and taking on responsibility as a caregiver [27]. While understanding public cooperation is required, our participants identified that repetitive messaging of health orders and restrictions with no positive projection for the future could easily lead to information and compliance fatigue and therefore threaten individual trust for the government.

While older adults may have more perceived emotional regulation strategies such as mature social networks that allow them to focus on emotions to reduce adverse effects, young adults may require active and ongoing exposure to positive messaging in order to generate positive attitudes and emotions [19, 28]. Most existing public health messaging focuses on the collective good using inclusive language, providing ideas to social safety, or emphasizing the risk of contracting COVID-19, which may elicit fear [29]. There is a reluctance to incorporate positive messaging because of potentially creating perceptions of false security, though available evidence does not support this concern and suggests compensation is not discernible at a population level [30]. Our data suggest not providing positive outlooks risk alienating and disengaging this age group linking to long-term pandemic response success.

Instilling hope during a public crisis is a powerful tool in combination with fear [29, 31]. An instillation of hope, besides mathematical modeling, should include the acknowledgment of the public's sacrifices and summarization of the public's response since the start of the pandemic to flatten the curve [32]. For example, Angela Merkel, Chancellor of Germany, used science and a clear explanation of the disease reproduction number and modeling to discuss an exit out of lockdown. While our participants agree it is often difficult to change behaviours of those who hold on tightly to their beliefs, an installation of hope through positively framed messaging may bring people together to facilitate collective change.

## The power of being heard

Almost all young adults who participated in focus group discussions were aware of media coverage on events caused by people in their age group that resulted in community spread. Yet, many raised questions on the accuracy and intentions of the reports as the biased wordings in many media messages framed young adults in a shameful way. Shaming is an inefficient method for behaviour change as it entrenches polarization, and discourages compliance [29]. On the other hand, messaging that shows an understanding of people's situations, sitting in someone else's shoes, effectively elicits compassion lead to corporations among the public [29]. During the time of the focus group, the province had been in a state of emergency for approximately eight months, with changing public health orders restricting social gatherings, traveling, and indoor activities. The province had been in full lockdown since early spring of 2020. While the provincial restrictions had relaxed for the summer of 2020 following the province's restart plan, the declaration of a second wave in October amended further strict regulations on gatherings. Furthermore, the provincial health officer issued two-week regional lockdown measures, including restricting all social gatherings of any size. While public services such as schools remained open, local small businesses such as restaurants and dance studios suspended operations according to changing health orders. A provincial mask mandate was issued shortly after the focus groups were conducted.

Young adults are more likely to rely on public transportation, work in essential service positions, and live in houses with multiple roommates that reduce cost but also increase exposure to the virus. To alleviate these unique group of stressors, providing an official platform for young adults to share their obstacles and lived experiences had been utilized in other nations. For example, in New Zealand Jacinda Ardern has live-streamed her conversations with regular New Zealanders to share their stories and advice in "Conversations through COVID". It is notable that numerous participants reached out to us following completion the study to express their gratitude for an opportunity to communicate their thoughts and feelings.

## Two-way communication

The varying interpretations and confusion around BC's public health orders was a reoccurring topic of conversation amongst our participants. Inconsistency between media reports and

public health reports was a persistent issue. For example, participants had questions understanding the change in definition of a "close contact" and were interested in knowing what they can do instead of what they cannot do. Following the change in health orders from "safe six" (six individuals one can contact with consistently besides the immediate family), to "core bubble" (maximum of two people one can regularly interact with if one lives alone), some participants were unaware of the change and for those who were aware, many questioned why in-person interactions such as at schools were still allowed. Public health research also found messages that came out first tend to be believed the most strongly [33]. Delays and uncertainties in understanding public health orders suggest improvements need to be made to increase the reach and effectiveness of public health messaging–regardless of the content of the message.

Our results indicate the mediums young adults use to actively seek for information is often different from the mediums they interact with the most. Therefore, the optimum strategy would be providing credible information on mediums that young adults interact with the most. Indeed, effective approaches to young adults in public health messaging have been complex in past health initiatives for communicable and chronic diseases including sexual health promotion, and the H1N1 outbreak [23, 34, 35]. However, tailored messages on a medium that is convenient, one that is highly accessible and widely used was found to be most effective in engaging youth [23, 36–38].

Social media was suggested as a frequently used medium and a widely accessible platform where young adults are comfortable sharing their concerns. Efforts have been made both locally and internationally to share credible information on social media, where users are perceived to be engaged in two-way communication [39]. The extended social networks that stem from social media can amplify positive behaviours, especially when central figures such as influencers spread positive health behaviours like handwashing and physical distancing. Previous health interventions had leveraged social media to deliver tailored messaging as young adults are mainly digital health users with high digital literacy [40]. Our participants had specifically mentioned following individuals that are active in science communication, such as Dr. Samantha Yammine, on social media. Similarly, other health care professionals have been active in delivering credible information on social media, such as Dr. Christian Drosten, a leading virologist who launched a podcast to explain the science behind the virus and the latest research [41].

### Strength and limitations

Strengths of this study include providing a unique young adult perspective by investigating the lived experiences of young adults using qualitative analytic strategies resulting in concrete themes for consideration when developing future public health messaging. Moreover, this paper suggests innovative ways to connect with this age group such as facilitating two-way communication using social media. All participants had some degree of post-secondary education which limited our ability to gain perspectives from individuals who are less educated or with limited language ability and technological access. Participants may also be influenced by social desirability factors, and withheld information if they failed to comply to public health orders.

### Conclusion

Our findings suggest messages that reach young adults should 1) have a positive framing, 2) reflects lived experiences of this age demographic, and 3) delivered on an accessible platform Contrary to popular belief, young adults perceive a high level of concern and stress regarding the pandemic to keep themselves and their loved ones safe. While young adults may have

questions and concerns towards public health messaging, a traditional broadcasting one-directional communication strategy makes voicing concerns difficult. Respectfully, we urge stakeholders including government officials and media outlets to head these recommendations, reporting and creating messaging that answers young adults' concerns. Tailored messaging is needed, desperately.

## Supporting information

**S1 Appendix.**
(DOCX)

## Acknowledgments

We thank the participants for their generosity with their time. SAL holds the Pfizer/Heart and Stroke Foundation Chair in Cardiovascular Prevention Research at St. Paul's Hospital

## Author Contributions

**Conceptualization:** Tina Cheng, Braxtyn Horbay, Lesley Lutes, Scott A. Lear.

**Data curation:** Tina Cheng, Rochelle Nocos.

**Formal analysis:** Tina Cheng, Braxtyn Horbay, Rochelle Nocos.

**Funding acquisition:** Scott A. Lear.

**Investigation:** Tina Cheng, Braxtyn Horbay, Rochelle Nocos.

**Methodology:** Tina Cheng, Braxtyn Horbay, Rochelle Nocos, Lesley Lutes, Scott A. Lear.

**Project administration:** Tina Cheng, Rochelle Nocos, Scott A. Lear.

**Supervision:** Lesley Lutes, Scott A. Lear.

**Writing – original draft:** Tina Cheng.

**Writing – review & editing:** Braxtyn Horbay, Rochelle Nocos, Lesley Lutes, Scott A. Lear.

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
