## [Decision Letter · Decision Letter 0]

11 Jun 2021

PONE-D-21-17314

The Role of Tailored Public Health Messaging to Young Adults during COVID-19: “There’s a lot of ambiguity around what it means to be safe”

PLOS ONE

Dear Dr. Lear,

Thank you for submitting your manuscript to PLOS ONE. After careful consideration, we feel that it has merit but does not fully meet PLOS ONE’s publication criteria as it currently stands. Therefore, we invite you to submit a revised version of the manuscript that addresses the points raised during the review process.

We look forward to receiving your revised manuscript.

Kind regards,

Prof. Anat Gesser-Edelsburg, Ph.D.

Academic Editor

PLOS ONE

Journal Requirements:

" We thank the participants for their generosity with their time. This research received no external

funding. SAL holds the Pfizer/Heart and Stroke Foundation Chair in Cardiovascular Prevention

Research at St. Paul’s Hospital."

Additional Editor Comments (if provided):

Reviewers' comments:

Reviewer's Responses to Questions

**Comments to the Author**

1. Is the manuscript technically sound, and do the data support the conclusions?

Reviewer #1: Yes

Reviewer #2: Yes

2. Has the statistical analysis been performed appropriately and rigorously? 

Reviewer #1: I Don't Know

Reviewer #2: N/A

3. Have the authors made all data underlying the findings in their manuscript fully available?

Reviewer #1: No

Reviewer #2: No

4. Is the manuscript presented in an intelligible fashion and written in standard English?

Reviewer #1: Yes

Reviewer #2: Yes

5. Review Comments to the Author

Reviewer #1: The authors have put together an interesting and relevant report especially with regard to matters affecting young adults during the COVID-19 pandemic. The manuscript is sound. Although it is usually expected that focus groups should comprise 6-8 participants, the the 4-6 participants in this study is still acceptable. Perhaps, the authors should replace the word 'stakeholds' with ' stakeholders' such as in the conclusion. Overall, this research is good, and should be published

Reviewer #2: This is a timely and very-well conducted and reported focus group study. In particular, the description of focus group methodology and coding process are exemplary. I have a couple of larger points and some minor stylistic points that should be addressed to further improve the manuscript, but have little doubt that the authors will be able to do so.

Major points:

1. I would like to see more of a justification for the use of focus groups. The manuscript nods to some drawbacks (like stronger social-desirability bias) and make use of some of the more interactive characteristics of the method, but I’d like to see more of a weighing of different options of collecting qualitative data and their suitability for the questions of interest (e.g., why were focus groups deemed a better model than semi-structured interviews?)

2. Especially in the discussion, the manuscript should be more circumspect about the generalizability of their findings – which is minimal. As noted under limitations, the sample is not representative and, given recruitment strategy, likely significantly biased towards those more informed (and likely more concerned) about COVID-19. For example, the study methodology simply doesn’t permit statements such as “Overall, we found young adults, contrary to mass media coverage, were highly concerned about the spread of COVID-19and perceived it as their responsibility to protect themselves and their loved ones.” (p. 14 – there are a number of other similar examples but this is highlighted as one of the study’s principal findings). Instead, I would encourage the authors to focus on the themes around messaging and perceptions (as a good bit of the discussion already does), where qualitative insights are of particular richness and value.

3. The authors’ use of “thematic analysis” does not correspond to Braun and Carke’s version of this (see their 2019 article [1], including the note that notions of intercoder reliability and multiple coders are irreconcilable with “reflexive thematic coding.” As the 2019 update helpfully clarifies, there are other, more positivist approaches to TA that are more appropriately referenced here.

Minor points:

1. It would be helpful to very briefly outline the COVID-19 situation and government response in Canada at the time of the FGs. That’s most relevant with respect to economic pressures faced by participants, but also to better understand their assessment of the government’s response and communication strategy

2. Figure 1 is a bit odd – I understand these are the four main themes, but does the way they’re arranged and connected have any meaning? Based on the text, I don’t think so. If they don’t, maybe just putting this in a table (perhaps even with a bit more definition) might be clearer

3. The percentage figures in table 1 are confusing. For the first column, they add up to 100% vertically, but for the 2nd and 3rd column they add up to 100% horizontally. I think just using the same logic across all columns makes most sense.

4. It’s not clear why there are two different formats for participant codes (e.g. p. 9 has “FG2 F” and “A109”.) Also not sure if using the codes adds anything over, say, using gender & age of the participant, but that’s just a though – just using a unified set of codes throughout is also fine.

5. The name of the virologist with a podcast (p. 18) is misspelled. It should be “Drosten”. I’d also consider citing the podcast as it is mentioned as an exemplary form of outreach.

Just as a note since I selected "no" for data sharing: The justification for not sharing the data is acceptable and the data availability statement phrased appropriately.

[1] Virginia Braun & Victoria Clarke (2019) Reflecting on reflexive thematic analysis, Qualitative Research in Sport, Exercise and Health, 11:4, 589-597, DOI: 10.1080/2159676X.2019.1628806

6. PLOS authors have the option to publish the peer review history of their article (what does this mean?). If published, this will include your full peer review and any attached files.

Reviewer #1: No

Reviewer #2: **Yes: **Sebastian Karcher

---

## [Author Response · Author response to Decision Letter 0]

23 Jul 2021

Please see attached Response to Reviewers' document.

---

## [Decision Letter · Decision Letter 1]

21 Sep 2021

The Role of Tailored Public Health Messaging to Young Adults during COVID-19: “There’s a lot of ambiguity around what it means to be safe”

PONE-D-21-17314R1

Dear Dr. Lear,

We’re pleased to inform you that your manuscript has been judged scientifically suitable for publication and will be formally accepted for publication once it meets all outstanding technical requirements.

Kind regards,

Prof. Anat Gesser-Edelsburg, Ph.D.

Academic Editor

PLOS ONE

Additional Editor Comments (optional):

Reviewers' comments:

Reviewer's Responses to Questions

**Comments to the Author**

1. If the authors have adequately addressed your comments raised in a previous round of review and you feel that this manuscript is now acceptable for publication, you may indicate that here to bypass the “Comments to the Author” section, enter your conflict of interest statement in the “Confidential to Editor” section, and submit your "Accept" recommendation.

Reviewer #2: All comments have been addressed

2. Is the manuscript technically sound, and do the data support the conclusions?

Reviewer #2: Yes

3. Has the statistical analysis been performed appropriately and rigorously? 

Reviewer #2: N/A

4. Have the authors made all data underlying the findings in their manuscript fully available?

Reviewer #2: No

5. Is the manuscript presented in an intelligible fashion and written in standard English?

Reviewer #2: Yes

6. Review Comments to the Author

Reviewer #2: The authors have effectively addressed all my comments. Congratulations!

(As a minor editorial note, Table 3 -- which is very effective -- uses "participant perceptions" and "participant's perception"; unless these are different things, I'd recommend using a single wording).

7. PLOS authors have the option to publish the peer review history of their article (what does this mean?). If published, this will include your full peer review and any attached files.

Reviewer #2: **Yes: **Sebastian Karcher

---

## [Editor Report · Acceptance letter]

24 Sep 2021

PONE-D-21-17314R1 

The Role of Tailored Public Health Messaging to Young Adults during COVID-19: “There’s a lot of ambiguity around what it means to be safe” 

Dear Dr. Lear:

I'm pleased to inform you that your manuscript has been deemed suitable for publication in PLOS ONE. Congratulations! Your manuscript is now with our production department. 

Kind regards, 

on behalf of

Prof. Anat Gesser-Edelsburg 

Academic Editor

PLOS ONE